# Osteoclast and Sclerostin Expression in Osteocytes in the Femoral Head with Risedronate Therapy in Patients with Hip Fractures: A Retrospective Comparative Study

**DOI:** 10.3390/medicina58111566

**Published:** 2022-10-31

**Authors:** Hwan-Hee Lee, Eun-Yong Choi, Hyun-Sik Jun, Young-Yul Kim

**Affiliations:** Department of Orthopedic Surgery, Daejeon St. Mary’s Hospital, College of Medicine, The Catholic University of Korea, Daejeon 34943, Korea

**Keywords:** femoral head, hip fracture, osteoclast, risedronate, sclerostin

## Abstract

*Background and Objectives*: The majority of research on the effects of osteoporosis drugs has measured the bone mineral density (BMD) of the spine and femur through dual-energy X-ray absorptiometry (DEXA) and compared and analyzed the effects of the drugs through changes in the BMD values. This study aims to compare osteoclast and sclerostin expression in osteocytes after risedronate therapy by obtaining femoral heads from patients with hip fractures. *Materials and Methods*: We obtained the femoral heads of 10 female patients (age: ≥65 years) who received risedronate therapy for at least 1 year through hip arthroplasty during 2019–2021 (risedronate group). Meanwhile, 10 patients who had never received osteoporosis treatment were selected as controls using propensity scores with age, body mass index, and bone density as covariates (control group). While the osteoclast count was evaluated using tartrate-resistant acid phosphatase (TRAP) staining, the sclerostin expression in osteocytes was assessed using immunohistochemistry. Moreover, Western blotting and polymerase chain reaction (PCR) were performed for receptor activation of nuclear factor kappa-Β ligand (RANKL), RANK, osteoprotegerin (OPG), sclerostin, and bone morphogenetic protein-2 (BMP2). *Results*: TRAP staining revealed significantly more TRAP-positive cells in the control group (131.75 ± 27.16/mm^2^) than in the risedronate group (28.00 ± 8.12/mm^2^). Moreover, sclerostin-positive osteocytes were expressed more in the control group (364.12 ± 28.12/mm^2^) than in the risedronate group (106.93 ± 12.85/mm^2^). Western blotting revealed that the expressions of RANKL, RANK, sclerostin, and BMP2 were higher in the control group than in the risedronate group (*p* < 0.05). Furthermore, RANK, sclerostin, and OPG protein levels were higher in the control group than in the risedronate group. *Conclusions*: In this study, the risedronate group demonstrated lower osteoclast activity and sclerostin expression in osteocytes in the femoral head than the control group.

## 1. Introduction

Bisphosphonate drug therapy is the first line of osteoporosis prevention and treatment, and many multinational, randomized controlled trials have reported a reduction in the incidence of hip fractures, one of the typical osteoporosis fractures, with bisphosphonate drug therapy [1,2,3,4]. Among them, risedronate is an oral bisphosphonate agent extensively used to treat osteoporosis, with proven efficacy in preventing the occurrence of fractures [5,6]. Particularly, risedronate is a third-generation bisphosphonate agent with potent anti-osteoclast activity [7,8] 

While significant research has been conducted on the treatment of risedronate in osteoporosis, most studies have compared the bone mineral density (BMD) values of the spine and femur through dual-energy X-ray absorptiometry (DEXA) to assess the treatment effects [5,9,10]. Of course, it has been reported that risedronate increases BMD levels and prevents osteoporotic fractures [5,6,11]. However, DEXA only provides information on BMD and does not reflect the effect of treatment, especially for osteoclasts [12,13]. Bone turnover markers are also widely used to compare treatment effects [14,15]. Although the bone turnover marker reflects osteoclast activity, it does not separately reflect areas where fractures occur frequently, such as the hip joint [16]. 

Recently, many studies have been conducted on the role of sclerostin and Wnt signaling in the treatment of osteoporosis [17,18]. However, there are still many uncertainties about the effect of sclerostin, and in particular, there are not many studies on the correlation between sclerostin and risedronate, including bisphosphonates. There is a study related to sclerostin and risedronate in mice [19], but as far as we know, there are no studies on bones from patients with fractures taking osteoporosis medications.

This study aims to directly examine the osteoclast and sclerostin expression in osteocytes in the femoral head removed during surgery to determine the cellular activity that can induce osteoporosis in the femoral head following risedronate treatment. To the best of our knowledge, there is no study comparing the effects of risedronate on the femoral head of patients with hip fractures. The findings of this study will enable direct comparative analysis rather than indirect comparison of the osteoporosis treatment’s effects through BMD values.

## 2. Materials and Methods

### 2.1. Specimen Collection

This retrospective comparative study was approved by the institutional review board of the Catholic University of Korea Daejeon St. Mary’s hospital (joint approval no. DC21SISI0076). We obtained written informed consent from all patients in this study. Between July 2019 and December 2021, we included female patients aged ≥65 years who had undergone hip arthroplasty for hip fractures. However, patients who were unsure about taking osteoporosis drugs or had taken them for <1 year and those who had taken osteoporosis drugs for >1 year but had taken ≥2 drugs were excluded from this study. Finally, 10 patients who received risedronate treatment (Actonel^®^ 35 mg, 150 mg) for at least 1 year were allocated to the risedronate group (Figure 1). Meanwhile, a matched comparison group (control group) was also created to decrease the effects of selection bias and potential confounding. Using propensity scores [20], we matched patients in the control group who did not receive osteoporosis treatment at all with each of the 10 patients in the risedronate group for age, body mass index (BMI), and hip BMD as covariates. We recorded patients’ age, BMI, hip BMD, and the American Society of Anesthesiologists (ASA) score. Furthermore, the treatment duration was examined in the risedronate group.

### 2.2. Histological Analysis of Decalcified Bone Tissue

The femoral head obtained through hip arthroplasty surgery was cut in the coronal plane using an oscillating saw (RuiJin Medical Instrument & Device Co., Ltd., Wuhu, China). The collected femoral head bones were fixed in 10% neutral buffered formalin for 24 h and later decalcified using 0.5 m ethylenediaminetetraacetic acid (EDTA, pH 7.4) for 4 weeks. The EDTA solution was replaced weekly until the transparent surface of the bone tissue was visible. After decalcification was completed, the bone tissue samples were dehydrated conventionally and embedded in paraffin wax, cut into 5-μm-thick sections, stained with hematoxylin and eosin (H&E), and analyzed microscopically (BX53; Olympus, Tokyo, Japan).

### 2.3. TRAP Staining

The tissue sections were deparaffinized using xylene, hydrated through graded alcohols, and placed in deionized water for 5 min. For tartrate-resistant acid phosphatase (TRAP) staining, we prepared TRAP basic incubation buffer with 1-L distilled water containing 9.2 g of sodium acetate anhydrous (Sigma S-2889; Sigma, St. Louis, MO, USA) and 11.9 g of L-(+) tartaric acid (Sigma T-6521; Sigma, St. Louis, MO, USA). For every 200 mL of TRAP staining solution, the TRAP basic incubation buffer was supplemented with 120 mg of Fast Red Violet LB salt (Sigma F-3381; Sigma, St. Louis, MO, USA) and 20 mg of Naphthol AS-MX Phosphate (Sigma N-4875; Sigma, St. Louis, MO, USA). Then, deparaffinized sections were incubated in the TRAP staining solution at 37 °C for 1 h until bright red TRAP was observed. After finishing the incubation, sections were rinsed three times in distilled water for 5 min and later counterstained with 0.02% Fast Green (Sigma-F7252; Sigma, St. Louis, MO, USA) for 30 s and rinsed five times in distilled water for 3 min. Next, sections were dehydrated quickly through graded alcohols, 5 s each, cleared in xylene, and mounted. We quantitated osteoclast numbers in the total region of the femoral head and calculated the ratio of the osteoclast numbers/bone surface (N/mm^2^). All values were calculated from at least 10 nonconsecutive sections per tissue. The images were obtained using an Olympus BX53 microscope (Olympus, Tokyo, Japan) equipped with a DP71 camera (Olympus, Tokyo, Japan).

### 2.4. Sclerostin Immunohistochemistry

The tissue sections were deparaffinized through xylene, hydrated through graded alcohols, and placed in deionized water for 5 min. For sclerostin immunostaining, we performed antigen retrieval using 0.05% trypsin solution for 20 min at 37 °C and incubated the sections in 3% hydroperoxide for 10 min. For the primary antibody, we used the mouse monoclonal anti-SOST antibody (sc-518161; 1:100, Santa Cruz, Dallas, TX, USA) at 4 °C overnight. Then, we used the VECTASTAIN Elite ABC kit (PK-6102; VECTOR Labs, Newark, NJ, USA) and ImmPACT NovaRED (SK-4805; VECTOR Labs, Newark, NJ, USA), per the manufacturer’s protocols. Next, cells were counterstained with Harris hematoxylin for 3 s and rinsed three times in distilled water for 5 min. Then, sections were dehydrated quickly through graded alcohols, 5 s each, cleared in xylene, and mounted. We quantitated sclerostin-positive cell numbers in the total region of the femoral head, expressed as sclerostin-positive cell numbers/bone surface (N/mm^2^). All values were calculated from at least 10 nonconsecutive sections per tissue. The images were obtained using an Olympus BX53 microscope (Olympus, Tokyo, Japan) equipped with a DP71 camera (Olympus, Tokyo, Japan). 

### 2.5. Western Blotting

The samples of femoral head tissues from a liquid nitrogen tank were mortared into powder. The powdered samples were washed in nuclease-free water three times to remove the remaining blood. Then, the total protein was extracted using RIPA buffer (Thermo Scientific, Maltham, MA, USA), supplemented with protease and phosphatase inhibitor, and the concentration of protein was measured per the Bradford dye-binding method (Bio-Rad, Hercules, CA, USA). Next, 20–50 µg protein were electrophoresed on 10%–15% sodium dodecyl sulphate–polyacrylamide gel electrophoresis (SDS–PAGE) gel and transferred to nitrocellulose membranes (GE Healthcare, Chicago, IL, USA). The probed membranes were blocked with 5% bovine serum albumin (BSA) solution for 1 h at 25 °C and incubated with primary antibody overnight at 4 °C. The primary antibodies used were as follows: receptor activator of nuclear factor kappa-Β (RANK; 1:1000, sc-374360; Santa Cruz, Dallas, TX, USA), receptor activator of nuclear factor kappa-Β ligand (RANKL; 1:1000, ab9957; Abcam, Cambridge, UK), SOST (1:1000, sc-518161; Santa Cruz, Dallas, TX, USA), bone morphogenetic protein-2 (BMP2; 1:1000, ab214821; Abcam, Cambridge, UK), and osteoprotegerin (OPG; 1:1000, ab73400; Abcam, Cambridge, UK). The membranes were washed three times with 1x Tris-buffered saline with Tween (TBST) solution, and the secondary antibody dissolved in 5% skimmed milk was probed at room temperature for 2 h. The secondary antibodies used were as follows: anti-rabbit immunoglobulin G (IgG; 1:5000, #7074) and ß-actin-HRP-conjugated (1:5000, #5125) were from Cell Signaling (Danvers, MA, USA), and goat anti-mouse IgG–HRP (1:5000, sc-2005) was from Santa Cruz (Dallas, TX, USA). The membranes were washed five times, and the protein activity was detected using enhanced chemiluminescence (ECL; Amersham ECL Prime Western Blotting Detection Reagent; GE Healthcare, Chicago, IL, USA). We calculated the quantification of protein expression using ImageJ version 1.53 (Wayne Rasband, Madison, WI, USA).

### 2.6. Quantitative Real-Time Polymerase Chain Reaction (PCR)

The samples of femoral head tissues from a liquid nitrogen tank were mortared into powder. The powdered samples were washed in nuclease-free water three times to remove the remaining blood. The total RNA was extracted from the bone powder using NucleoZOL (Macherey-Nagel Gmbh & Co. KG, Dueren, Germany) per the manufacturer’s protocols. Then, cDNA was synthesized using 0.5 µg total RNA with a ReverTra Ace qPCR RT Master Mix kit (TOYOBO, Osaka, Japan), per the manufacturer’s protocols. The synthesized cDNA was amplified with the target gene primers and THUNDERBIRD SYBR qPCR Mix (TOYOBO) using the 7500 Fast Real-Time PCR System (Applied Biosystems, Maltham, MA, USA). The amplification conditions were as follows: denaturation at 95 °C for 10 min, amplification with 40 repeated cycles at 95 °C for 15 s and 60 °C for 1 min, and melting curve analysis at 95 °C for 15 s, 60 °C for 1 min, and 95 °C for 30 s. The relative gene expression data were normalized to GAPDH gene expression. Table 1 lists the primers (Table 1).

### 2.7. Statistical Analysis

All statistical analyses were performed using GraphPad Prism software (GraphPad Software ver.5.01, Chicago, IL, USA) and SPSS software ver. 21.0 (SPSS, Chicago, IL, USA). All data are presented as mean ± standard deviation. Since the number of samples was small, a normality test (Shapiro–Wilk) was performed and satisfied. While continuous variables were compared using the unpaired Student’s *t*-test or the Mann–Whitney test, dichotomous variables were compared using the chi-square test or Fisher’s exact test (as appropriate). In addition, propensity score matching was generated from a logistic regression using the covariates mentioned above. In this study, *p* < 0.05 was considered statistically significant.

## 3. Results

### 3.1. Specimen Collection

In the risedronate group, the mean age of the study participants was 81.1 ± 5.9 years, the mean BMI was 22.74 ± 2.58, the mean hip BMD was −3.06 ± 0.82, and the mean ASA score was 1.9 ± 0.74. In the control group, the mean age was 82.2 ± 4.1 years, the mean BMI was 23.19 ± 4.03, the mean hip BMD was −3.19 ± 0.84, and the mean ASA score was 2.0 ± 0.82. We observed no statistically significant difference between age, BMI, hip BMD, and ASA scores in both groups (Table 2). Furthermore, the average treatment period of the risedronate group was 23.7 ± 15.9 months.

### 3.2. TRAP Staining

TRAP-positive osteoclasts were present along the trabecular surface, and resorption bays were confirmed in some tissue slides (Figure 2A). Owing to counting TRAP-positive osteoclasts in the entire femoral head, an average of 131.75 ± 27.16/mm^2^ was established in the control group and 28.00 ± 8.12/mm^2^ in the risedronate group (Figure 2B). We observed significant differences in both groups (*p* < 0.005).

### 3.3. Sclerostin Immunohistochemistry

Osteocyte activity was established through sclerostin. In addition, many osteocytes were identified in the bone trabeculae; some osteocytes showed intense labeling for sclerostin, while others displayed weak staining of sclerostin (Figure 2C). When intensely labeled osteocytes with sclerostin were counted in the femoral head, an average of 364.12 ± 28.12/mm^2^ was observed in the control group, whereas only a small number was established in the risedronate group (106.93 ± 12.85/mm^2^; Figure 2D). A statistically significant difference was noted between both groups (*p* < 0.005).

### 3.4. Western Blotting and Quantitative Real-Time PCR

The expressions of RANK, RANKL, SOST, BMP2, and OPG were visualized by Western blotting. The quantitative results revealed that the expression of RANK, RANKL, SOST, and BMP2 was higher in the control group than in the risedronate group. Moreover, OPG was higher in the risedronate group than in the control group (Figure 3). We observed significant differences in both groups (*p* < 0.005).

In addition, qPCR showed that the expression of RANK and SOST was higher in the control group than in the risedronate group. While OPG was higher in the risedronate group than in the control group (*p* < 0.05), the expression of RANKL and BMP2 was higher in the control group than in the risedronate group but with no statistically significant difference (Figure 4).

## 4. Discussion

In our study, by establishing the osteoclast activity of the femoral head through TRAP staining, much less TRAP (+) osteoclast was confirmed in the risedronate group than in the control group. In addition, Western blotting and qPCR demonstrated that both RANKL and RANK associated with the osteoclast activity were expressed less in the risedronate group than in the control group, thereby supporting the same results as the abovementioned TRAP staining results. These findings also show that risedronate effectively inhibits osteoclasts in the femoral head.

Another interesting result of our study was that of sclerostin. We confirmed that the number of intense labeling osteocytes with sclerostin was much lower in the risedronate group than in the control group. Furthermore, Western blotting and PCR confirmed that the sclerostin expression was much less in the risedronate group, thereby supporting the abovementioned results.

Sclerostin is a protein expressed in the SOST gene [21]. In adult bone, sclerostin is expressed only by osteocytes and functions to inhibit bone formation by osteoblasts [22,23]. Sclerostin inhibits canonical Wingless (Wnt)/ß-catenin signaling by binding to low-density lipoprotein receptor-related proteins (LRP)-5 and LRP-6 [24,25]. To date, limited studies have reported the correlation between sclerostin and risedronate, including bisphosphonate. 

Morales-Santana et al. [26] reported that serum sclerostin was high in inadequate responders who had fractures among patients treated with bisphosphonate in postmenopausal women and that high serum sclerostin may be one of the factors that can predict the risk of fracture. Polyzos et al. [27] reported an increase in serum sclerostin levels when risedronate was administered to postmenopausal women for 6 months; however, they did not explain the results and the mechanism of the correlation between risedronate and sclerostin. Our results demonstrated that the number of intense labeling osteocytes with sclerostin in the bone was much lower in the risedronate group. In the study by Polyzos et al. [27], the average age of examined patients was 64.2 years, which was approximately 16 years younger than our study sample, the administration period of risedronate was marginally shorter at 6 months, and they measured serum sclerostin. In our study, osteocytes with intense activity were counted by immunochemical staining of sclerostin in the bone, not serum, using only the femoral head with fracture, and Western blotting and PCR presented the same results. Furthermore, Boltenstål et al. [28] reported no statistically significant correlation between serum sclerostin and sclerostin in the bone but rather a correlation with serum phosphate. 

In our study, no clear theoretical basis exists for the reduced sclerostin expression in the bone trabeculae in the risedronate group compared with the control group. However, as risedronate inhibits osteoclast activation, the bone remodeling cycle is decreased, which, in turn, decreases osteoblast activation; in this process, it could also be assumed that the osteocyte activity with the sclerostin expression is also reduced. In Western blotting, as the BMP2 expression was lower in the risedronate group than in the control group, it could serve as a basis for the low osteoblast activity in the risedronate group. Ozaki et al. [19] conducted an experiment on mandibles of OPG-deficient mice and reported that trap-positive multinucleated cells were less in mice administered with the risedronate and WP9QY, RANKL-binding peptides. In particular, in mice administered with WP9QY, more osterix-positive cells were expressed, and fewer sclerostin-positive osteocytes were expressed. Based on this, it was reported that WP9QY not only inhibits osteoclastogenesis but also improves osteoblastogenesis.

According to Chekroun et al. [29], matrix stiffness activates the WNT-catenin pathway to initiate a cascade of transcription factor production, leading to the production and exocytosis of mineralization enhancers and inhibitors. They proposed theoretical evidence for osteoblast self-inhibition after the activation of genetic regulatory networks without the action of osteoclasts. According to their theory, apart from risedronate, which inhibits osteoclasts, sclerostin inhibits the Wnt-catenin pathway and reduces the secretion of bone sialoprotein, Alkaline Phosphatase (ALP), osteocalcin (OC), and osteopontin (OPN) [29]. Although the level of ALP or OC was not measured in this study, it was confirmed that BMP2 appeared higher in the control group than in the risedronate group. As a result, it is possible that the osteoblast-expressed sclerostin in the control group was increased due to osteoblast self-inhibition. However, as the effect of osteoclast is excluded from this theory, further research is needed to support this hypothesis.

From another viewpoint, in femoral heads that did not receive treatment for osteoporosis, the activity of osteoclast and sclerostin increased at the osteocyte level. Therefore, the process of osteoporosis might increase not only osteoclast activity but also sclerostin, an antagonist of osteoblasts. Nevertheless, further studies are warranted on the relationship and related mechanisms of osteoclast activity and sclerostin activity in bone during osteoporosis. Furthermore, our study found no significant difference in hip BMD levels between both groups, but the osteoclast activity showed a significant difference. While hip fractures occurred in both groups, it is also meaningful to directly confirm that the osteoclast activity in the femoral head with similar BMD levels was lower in the risedronate group. 

Recently, many studies on mechanics and biology have been conducted through computer simulations [30,31]. In particular, in silico experiments have been used to elucidate the mechanobiology and signaling pathways of cells in clinical biopsies and animal bones, including the effects of drugs on single cells and signaling pathways [32]. Therefore, it is expected that a more accurate and larger amount of data can be easily obtained if future designs are planned through computer simulation and in silico research, in addition to the method of collecting the femoral head directly from the patient as in this study.

This study has some limitations worth acknowledging. First, this is a retrospective study. Second, this study did not follow up with individual patients continuously and compare their results but divided femoral heads with different individual characteristics into two groups and compared the results. Although we have attempted to compare both groups under the same conditions as much as possible without significant differences in age, BMI, hip BMD, and ASA scores, these limitations could not be avoided owing to differences in bone structure or individual bone quality. Moreover, because of the study design and methods, there was only one surgically removed femoral head; thus, there is no choice but to have a methodological limitation that one patient cannot be continuously followed. Third, the number of patients in each group was small. Among patients who underwent hip arthroplasty for hip fracture in our hospital, including many patients in this study was challenging because few patients satisfied the inclusion criteria. Furthermore, it was hard to increase the number of patients because it was crucial to match patients with as similar conditions as possible as a control group. Finally, the results of the serum bone marker were not presented. In the future, it is essential to examine the correlation between serum bone markers and osteoclasts or osteoblasts by sampling the femoral head over a long period in a prospective design.

## 5. Conclusions

This study found that the number of osteoclasts was lower in patients taking risedronate in the femoral head with hip fracture. Moreover, the activity of osteocytes with sclerostin in the femoral head was lower in risedronate-treated patients than in the control group. We believe that additional prospective studies involving more samples are required in relation to this research topic.

## Figures and Tables

**Figure 1 medicina-58-01566-f001:**
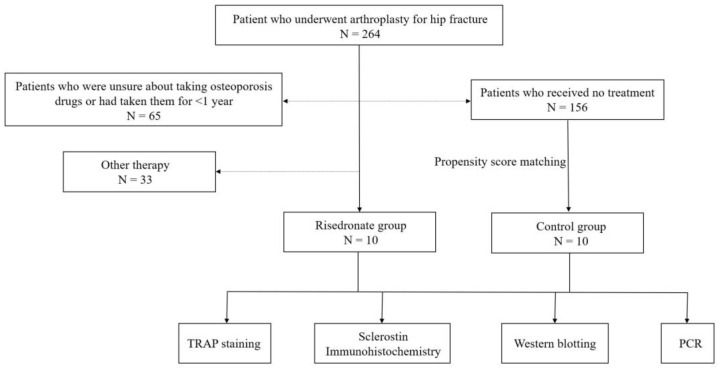
Flow chart of the study.

**Figure 2 medicina-58-01566-f002:**
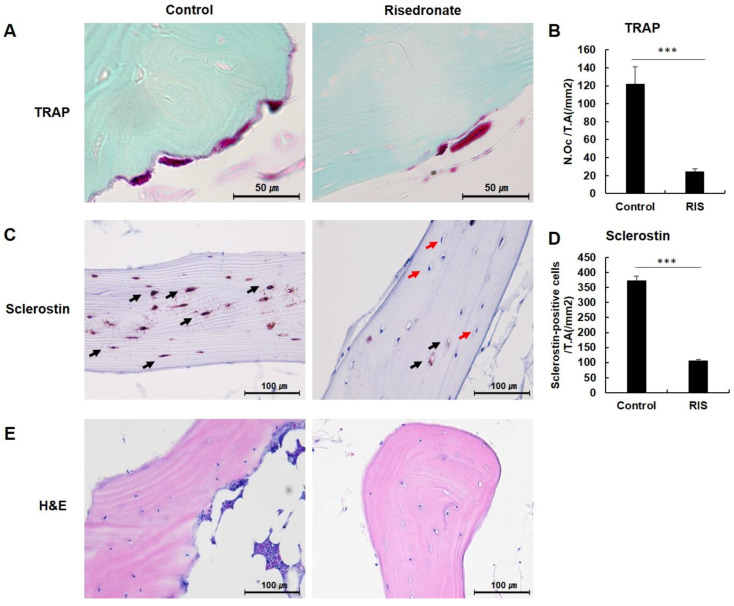
(**A**) Histology of tartrate-resistant acid phosphatase (TRAP) staining of the femoral head. (**B**) Quantitative analysis of the number of osteoclasts. (**C**) Sclerostin immunohistochemistry of the femoral head. Osteocytes show labeling for sclerostin (black arrow), and some osteocytes do not show staining of sclerostin (red arrow). (**D**) Quantitative analysis of the number of sclerostin-positive osteocytes. (**E**) Histology of hematoxylin and eosin (H&E) staining (*** *p* < 0.005).

**Figure 3 medicina-58-01566-f003:**
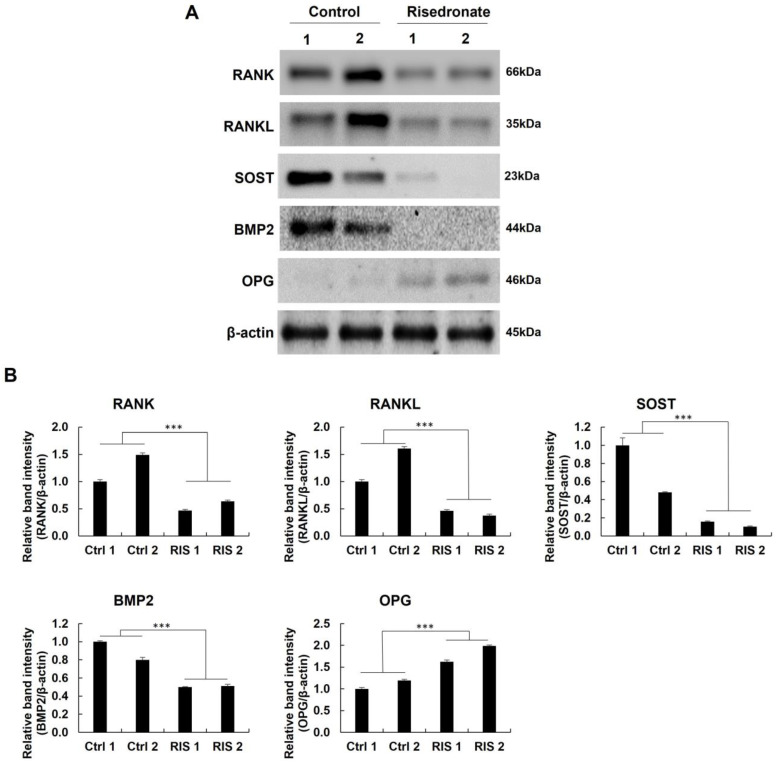
Western blotting results. (**A**) Band expression of RANK, RANKL, SOST, BMP2, and OPG. (**B**) Quantitative results of Western blotting for the expression of RANK, RANKL, SOST, BMP2, and OPG (*** *p* < 0.005).

**Figure 4 medicina-58-01566-f004:**
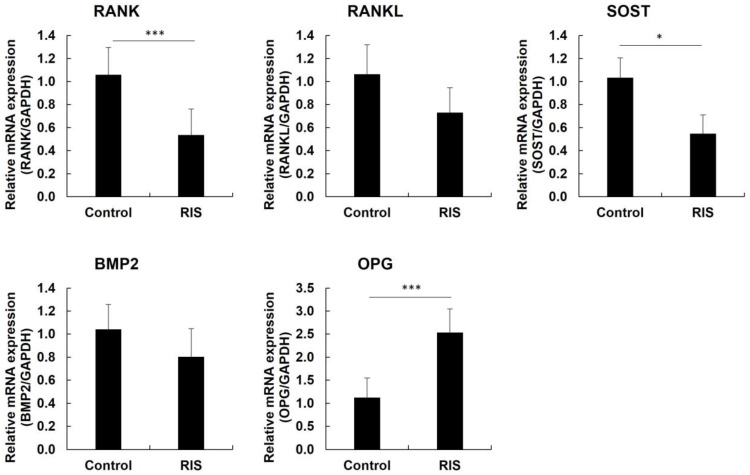
PCR results of RANK, RANKL, SOST, BMP2, and OPG (* *p* < 0.05, *** *p* < 0.005).

**Table 1 medicina-58-01566-t001:** PCR primers used in this study.

Primer	Direction	Sequence
GAPDH GAPDH RANK RANK RANKL RANKL SOST SOST BMP2 BMP2 OPG OPG	Forward Reverse Forward Reverse Forward Reverse Forward Reverse Forward Reverse Forward Reverse	CCTGCCAAATATGATGACATCAAG GTGGTCGTTGAGGGCAATG ACGTGGACCCTTGCCCCAGT ACTGGCCACCAGGGGAGCTT CACCATCAGCTGAAGATAGT CCAAGATCTCTAACATGACG AGACCAAAGACGTGTCCGAG GGGATGCAGCGGAAGTC ATGGATTCGTGGTGGAAGTG GTGGAGTTCAGATGATCAGC GTAGGTGCCAGGAGCCATT CAATGAACAAGTGGCTGTGC

PCR, polymerase chain reaction.

**Table 2 medicina-58-01566-t002:** Comparison of the control group and risedronate group.

	Control Group (*n* = 10)	Risedronate Group (*n* = 10)	*p*-Value
**Age (years)** **BMI (kg/m^2^)** **Hip BMD** **ASA score**	82.2 ± 4.1 23.19 ± 4.03 −3.19 ± 0.84 2.0 ± 0.82	81.1 ± 5.9 22.74 ± 2.58 −3.06 ± 0.82 1.9 ± 0.74	0.471 0.214 0.965 0.789

BMI, body mass index; BMD, bone mineral density; ASA, American Society of Anesthesiologists.

## Data Availability

Data available on request.

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
