# Peer review of "Osteoclast and Sclerostin Expression in Osteocytes in the Femoral Head with Risedronate Therapy in Patients with Hip Fractures: A Retrospective Comparative Study"

_medicina, 2022, doi:10.3390/medicina58111566_

Round 1
Reviewer 1 Report
It is a retrospective study that shows that the number of osteoclasts and activity of osteocytes with sclerostin was lower in patients taking RisedronateRisedronate in the femoral head with hip fracture.
This study does not bring much new information since RisedronateRisedronate is known for inhibiting osteoclastic activity and inducing apoptosis of osteoclasts, and adequate responders to biphosphonates show a decrease in sclerostin. (1) However, the level of response sclerostin to Risedronate is not established (2). It should be the main focus of the paper.
Authors present that when intensely labeled osteocytes with sclerostin were counted in the femoral head, an average of 364.12 209 ± 8.12/mm² was observed in the control group, whereas only a small number was established in the risedronate group (106.93 ± 2.85/mm²; Fig. 1D). A statistically significant difference was noted between both groups.
Please perform a power sample analysis to show the risk of a type I error. Please provide the beta of the study, and if it does not reach 0.8, please provide the number of samples needed to reach significance at that power. Please do the same for RANK, RANKL, SOST, BMP2, and OPG.
In order to decipher the mechanism of action of sclerostin and its interaction with other bone markers, please use the following references to enhance your discussion. For instance, expression of sclerostin, an inhibitor of Wnt/β-catenin signaling, was significantly lower in tibiae of OPG–/–mice than in wild-type mice. (3) In order to discuss the inhibition of the WNT pathway, please use the description of the WNT pathway in regards to mineralization enhancers and inhibitors as previously described and specifically in terms of self-inihibition (4) and the possible role of osteomirs.
(1) Morales-Santana S, Díez-Pérez A, Olmos JM, Nogués X, Sosa M, Díaz-Curiel M, Pérez-Castrillón JL, Pérez-Cano R, Torrijos A, Jodar E, Rio LD, Caeiro-Rey JR, Reyes-García R, García-Fontana B, González-Macías J, Muñoz-Torres M. Circulating sclerostin and estradiol levels are associated with inadequate response to bisphosphonates in postmenopausal women with osteoporosis. Maturitas. 2015 Dec;82(4):402-10. doi: 10.1016/j.maturitas.2015.08.007. Epub 2015 Aug 20. PMID: 26358930.
(2) Polyzos SA, Anastasilakis AD, Bratengeier C, Woloszczuk W, Papatheodorou A, Terpos E. Serum sclerostin levels positively correlate with lumbar spinal bone mineral density in postmenopausal women--the six-month effect of Risedronate and teriparatide.
Osteoporos Int. 2012 Mar;23(3):1171-6. doi: 10.1007/s00198-010-1525-6. Epub 2011 Jan 11. PMID: 21305266.
(3) Ozaki Y, Koide M, Furuya Y, Ninomiya T, Yasuda H, Nakamura M, Kobayashi Y, Takahashi N, Yoshinari N, Udagawa N. Treatment of OPG-deficient mice with WP9QY, a RANKL-binding peptide, recover alveolar bone loss by suppressing osteoclastogenesis and enhancing osteoblastogenesis. PLoS One. 2017 Sep 22;12(9):e0184904. doi: 10.1371/journal.pone.0184904. PMID: 28937990; PMCID: PMC5609750.
(4) Chekroun A, Pujo-Menjouet L, Falcoz S, Tsuen K, Yueh-Hsun Yang K, Berteau JP. Theoretical evidence of osteoblast self-inhibition after activation of the genetic regulatory network controlling mineralization.
J Theor Biol. 2022 Mar 21;537:111005. doi: 10.1016/j.jtbi.2022.111005. Epub 2022 Jan 12. PMID: 35031309.
Author Response
Thank you very much for your consideration. Responses to reviewers are sent as attachments.

Reviewer 2 Report
1. In the abstract section, quantitative findings should be reported.
2. Keywords should be reorganized alphabetically.
3. The Reviewer do not see the novel in the present article. My examination revealed that several similar previous publications appear to appropriately address the issues you have brought up in the current submission. Please emphasize it more advance in the introduction section if there are any more truly something really new.
4. To underline the study gaps that the newest research tries to fill, it is crucial to explain the merits, novelty, and limits of earlier studies in the introduction.
5. To make the content in the present article more comprehensive, the authors should discuss computational simulation/in silico study for research related to osteoporosis and bone mineral density. The introduction and/or discussion part of an article should contain this crucial topic, according to the authors. In addition, to reinforce this explanation, the recommended reference should be cited as follows: Ammarullah, M. I.; Santoso, G.; Sugiharto, S.; Supriyono, T.; Kurdi, O.; Tauviqirrahman, M.; Winarni, T. I.; Jamari, J. Tresca Stress Study of CoCrMo-on-CoCrMo Bearings Based on Body Mass Index Using 2D Computational Model. Jurnal Tribologi 2022, 33, 31–8. https://jurnaltribologi.mytribos.org/v33/JT-33-31-38.pdf
6. In order to improve the reader's understanding of the materials and methods section simpler, the authors could provide a figures that clarify the workflow of the current study rather than only the predominant text as it currently appears.
7. What is the basis for patient selection? Is there any protocol, standard, or basis that has been followed? It is unclear since the patient is very heterogeneous with a small number. The resonance involved impacts the present result makes this study flaws. One major reason for rejecting this paper.
8. It's also essential to include additional information on the manufacturer, country, and specifications of the tools.
9. A comparative assessment with similar previous research is required.
10. Before moving on to the conclusion section, the present study's limitations must be included.
11. The conclusion section needs to explain further research.
12. The authors need to enrich the reference from five years back.
13. Throughout the manuscript, the authors created paragraphs that were only one or two phrases long, making the explanation difficult to understand. The authors should expand on their explanation to make it a more thorough paragraph. It is advised that one paragraph have at least three sentences, with one sentence functioning as the primary sentence and the other sentences functioning as supporting sentences. See line 48-51.
14. Due to grammatical mistakes and English style, English has to be proofread.
15. Please be aware that the authors followed the MDPI format correctly; modify the current form and recheck, as well as any other problems that have been highlighted.
Author Response

(The authors gave the same response as above.)

Round 2
Reviewer 1 Report
The authors made the changes required. However, they must explain how their findings act on the biological content. Indeed, they used the citations suggested not to enrich the theoretical concept but copy and paste the results. It is not the sense of a discussion.
For instance, regarding the citations of Chekroun et al., "Through this, they proposed theoretical evidence for osteoblast self-inhibition after the activation of genetic regulatory networks that control mineralization." How does the self-inhibition of mineralization is linked to the use of ris? Inhibiting the Wnt/β-catenin should inhibit the secretion of BSP-ALP-OC and OPN (Chekroun et al.), decreasing the mineralization of the bone matrix. Did you find this? How are your results linked to this? Please elaborate on the link between a decrease in Sclerostin's impact on BMP2, OPG, and the WNT pathway. Same for the use of other references is the main weakness of the paper.
In addition, please remove the "perhaps" and other colloquial wordings. A native editor will bring substantial help on how to make it flawless.
Reviewer 2 Report
I am recommending the present work for accepted in the present form.
Author Response
We sincerely appreciate your acceptance.